# Application of Extended Finite Element Method for Studying Crack Propagation of Welded Strip Steel in the Cold Rolling Process

**DOI:** 10.3390/ma16175870

**Published:** 2023-08-28

**Authors:** Jianjun Chen, Chaojie Wu, Jiacong Ying

**Affiliations:** School of Mechanical and Power Engineering, East China University of Science and Technology, Shanghai 200237, China; 18579203417@163.com (C.W.); yingjiacong@163.com (J.Y.)

**Keywords:** extended finite element method, crack propagation, welded strip steel, cold rolling, numerical simulation

## Abstract

In the cold rolling process, edge cracks, particularly those near the welded zone, can inadvertently lead to strip rupture. This study employed the extended finite element method (XFEM) to analyze the crack propagation behavior in welded strip steel during cold rolling. Various tests such as the tensile test, essential work of fracture (EWF) test, spherical indentation method, and elastoplastic finite element simulations were conducted to determine the maximum principal stress and fracture energy utilized in XFEM for the base metal and weld metal, respectively. A continuous cold rolling model was established to investigate the crack propagation behaviors in the base metal, weld metal, and the interface between the base and weld metal. In the continuous rolling process, the crack propagation and expansion speed in the base metal are much larger than that of the weld zone. In addition, the base metal at the back end of the rolled piece is more prone to fracture than the base metal at the front end.

## 1. Introduction

Cold rolling is an important production process affecting the quality and efficiency of strip steel fabrication [1]. Before the strip steel enters into the cold rolling mill, the cutting operation would be carried out to remove the edge defects to ensure the strip steel integrity. However, a cutting operation cannot eliminate all edge defects and sometimes it will introduce new edge flaws due to the inappropriate cutting parameters [2]. These initial defects may cause edge cracking or even a strip steel rupture during the cold rolling process, which will lead the halt of whole production line. Previous studies showed that the edge cracking or strip rupture easy occur near the weld zone due to the complexity of the material combination [3]. How to avoid edge cracking and strip rupture in the cold rolling process has been a crucial factor in cold rolling.

Research on the causes of cold-rolled edge cracks and corresponding protective measures has achieved some results. The main reasons for the edge cracks of cold-rolled sheet are as follows: edge defects of cold-rolled substrate, uneven structure of cold-rolled substrate and cold-rolling process control problems, cold-rolling mill load distribution and improper control of tension between stands, discordant deformation of cold rolled sheets [4], and trimming the amount of edges [5].

With the assistance of the finite element method, researchers have studied different fracture behaviors of strip steel under cold and hot rolling condition. Compared with other numerical methods, the finite element method has distinct advantages especially for arbitrary geometric shape, boundary condition, and geometric nonlinearity. Several scholars have studied the crack surface morphology based on the extended finite element method. Yu et al. [6] analyzed the deformation behavior and crack initiation when the strip steel containing inner non-metallic inclusions under rolling condition by using the three–dimensional finite element method and geometric correction method. The relationship between deformation of inclusion and crack propagation was discussed. Ervasti and Ståhlberg [7] used finite element program to analyzed the behavior of longitudinal surface cracks and transversal cracks in hot rolling. The purpose of their study is to investigate the possibility of cracks disappearing during the cold rolling by controlling plastic deformation. Son et al. [8] investigated the closure and propagation behavior of surface defects with different initial sizes and locations in wire rod rolling. They found when the initial size and position of the intentionally created notch in the billet changes, the notch would shrink or widen. However, most of the above work were only conducted by elastic-plastic material model and can’t reveal the crack propagation behavior under the cold rolling process, all of which limit these researches’ instructive significance.

In the past, some researchers started to use continuum damage models to investigate edge crack behavior in cold rolling processes. Na and Lee [9] discussed the influence of roll unevenness on the initiation and propagation of edge cracks based on the strain control damage criterion. Ghosh et al. [10] applied the two damage models of Gurson–Tvergaard–Needleman (GTN) as well as Cockcroft-Latham to the prediction of edge cracking of cold-rolled strip steel, and found that these two damage models can achieve the effect of damage gathering at the edge of the strip. This is consistent with the possibility of cracking at the edges of cold-rolled strip steel in actual production. After analyzing the distribution of the stress and strain field at the tip of the edge defect, Yan et al. [11,12] used the GTN damage model to study the initiation and propagation of the edge crack. Through factorial analysis and orthogonal test, the influence of different process parameters on edge crack behavior was analyzed and a fracture criterion was proposed for the cold rolling. Sun et al. [13] analyzed the damage evolution and distribution of strip steel during rolling process based on the modified GTN damage model. Although above-mentioned research largely widens the fracture knowledge of the strip steel in cold rolling, these researches only considered the crack behavior in the base metal and the information about the weldment fracture under the cold rolling condition is still limited.

Belytschko and Black [14] came up with a minimal remeshing finite element method for crack propagation. This method allows the crack to be arbitrarily aligned within the mesh. But for severely curved cracks, remeshing may be needed. Using the conventional finite element method has some shortcomings to simulate the crack propagation. In 1999, Nicolas and Belytschko [15] proposed an extended finite element method (XFEM) to solve the discontinuities in the conventional finite element framework. The extended finite element method is a good way to analyze the difficulties encountered by the conventional finite element method for solving the problem of crack propagation [16]. This method is based on the element decomposition in which the finite element meshes and the cracks are independent from each other. When the crack extends, the model needn’t be remeshed continually. After years of development, XFEM has been widely applied in dealing with crack propagation problems. Wells et al. [17] used XFEM to solve the adhesive crack problem in quasi-brittle heterogeneous materials. They used a cohesive crack model for numerical simulations. It has shown that this method can objectively simulate cracks propagation with unstructured mesh. Nicolas and Belytschko [18] used XFEM to simulate the propagation path of arbitrary crack types in the concrete. The simulation results of concrete bond crack propagation showed the effectiveness of the proposed method. These researches indicated that XFEM is a convenient way to study the fracture problem.

In recent research, Tian Jing [19] conducts a systematic study on the crack suppression mechanism of rolling under different thickness reductions. Jain, Vipul [20] investigated the role of microstructure, crystallographic texture, and analytical stress state on the formation of alligator crack and edge crack during cold rolling of Fe-3.78 wt.% Si electrical steel. Huang, Zhiquan [21] analyzed the influence of the rolling speed on the flow field and solidification welding line of AZ31 magnesium alloy in the roll casting area by finite element simulation. Xu, Wei [22] studied the effect of Al-Si coating on the weld bead formation, aluminum (Al) element distribution, the microstructure and mechanical properties of laser welded 22 MnB5 steel joint under various welding speed. WEI, Lianfeng [23] studied the effects of temperature and hydride on the fatigue crack propagation behavior of welded sheets. Sun Xian [24] studied the influence of weld metal on the sensitivity of austenitic stainless steel to the crack propagation. Most of the research focuses on the crack formation mechanism of materials, the prevention of cracks and weld zone properties. There are few studies on strip steel with welds zone.

In our previous research work, numerical simulation work was carried out on cold-rolled strip steel with prefabricated defects, which was verified by rolling tests. The results show that the simulation crack propagation path is similar to the rolling test [25]. In this study, the XFEM was used to analyze the edge cracking behavior of the weld-containing strip steel under the cold rolling condition. The mechanical parameters used in XFEM for the base metal and weld metal were determined, respectively, by the tensile test, the essential work of fracture (EWF) test, the spherical indentation method and the elastoplastic finite element simulation. A systematic analysis was performed to reveal the effects of defect location on the edge crack propagation behavior.

## 2. Material and Width of Welded Joints

The silicon strip steel sheet containing weldment was employed in this research work. The compositions of its chemical elements are listed in Table 1.

The cross-section of the welded strip steel was observed by metallographic microscope. The metallography along the width direction and the depth direction are illustrated in Figure 1. The width of the weld is about 1.5 mm at the top and 0.5 mm at the bottom.

Microhardness analysis was performed using the HX-1000TM microVickers hardness tester, manufactured by BAHENS INSTRUMENT CO.,LTD. in Shanghai, China, as shown in the Figure 2a. A load of 100 g and a dwell time of 15 s were used during the test. Figure 2b shows the inlaid sample of the strip steel with weldment. Two paths, Path1 and Path2, were taken into account in the microhardness test, where Path1 is 0.4 mm away from the top and Path2 is 0.3 mm away from the bottom, and take a sample point every 0.5 mm in the base metal zone and every 0.1 mm in the heat-affected zone and weld zone along the path.

The variation of that microhardness along Path1 and Path2 are shown in Figure 3. The microhardness value of the weldment is about 130–230 HV and the hardness of base metal is about 130 HV.

By measurement, the width of the upper weld zone of Path1 is approximately 1.4 mm, and the width of the upper weld zone of Path2 is approximately 0.7 mm. The interval widths of the base metal and the weld obtained from the microhardness test are consistent with the results observed in the previous metallographic test, which verifies the accuracy of the interval widths.

## 3. Mechanical Behavior of Weld Metal and Base Material

The tensile specimens of the base metal and weld metal were cut from the strip steel, as shown in Figure 4. The specimen meets the standard of ISO 6892.1-2009 which is shown in Figure 5a,b.

As shown in Figure 6a,b, tensile tests of base metal and weld material were carried out on Instron tensile testing machine and micro-tensile testing machine, respectively.

Figure 7 shows the stress-strain curve of the base metal and weld material obtained by tensile test. The material mechanical properties of base metal were obtained as: the elasticity modulus *E* = 220 GPa, the true tensile stress $\sigma_{b}$ = 410 MPa, the engineering tensile stress, Rm = 312 MPa and yield stress $\sigma_{s}$ = 238 MPa. The material mechanical properties of weld material were obtained as: the elasticity modulus *E* = 197 GPa, the true tensile stress $\sigma_{b}$ = 584 MPa, the engineering tensile stress, Rm = 534 MPa and yield stress $\sigma_{s}$ = 452 MPa. Meng et al. [26] studied the laser welding process. In the tensile test, they found that all of the specimens fractured at the base material away from the heat-affected zone and the fusion zone. This indicates that the strength of the weld zone is higher than that of the base material. The welding method used in this research is laser welding. The strength of the electrode is higher than the strength of the base metal. Therefore, the weld strength is higher than that of the base metal, and the yield strength is about twice that of the base metal.

In the tensile test, the relationship between tension and elongation before necking of the specimen can be measured accurately. The nominal and real stress strain curves are shown: the conversion formula of the real stress-strain and the nominal stress strain is as follows:(1)$$\varepsilon_{true}=\ln\left(1+\varepsilon_{nom}\right)$$
(2)$$\sigma_{true}=\sigma_{nom}\left(1+\varepsilon_{nom}\right)$$

In the Equation, $\varepsilon_{true}$ and $\varepsilon_{nom}$ are the true strain and nominal strain, respectively.

$\sigma_{true}$ and $\sigma_{nom}$ are the true stress and nominal stress, respectively. Through the conversion of the Equations (1) and (2), the true stress and strain relationship can be obtained. The true stress and true strain data can be input into the ABAQUS as the material parameters before the necking.

After the specimen is necked, the true stress relationship cannot be obtained from the tensile test. By means of the weighted mean Equation (3), the w is constantly changed and the relationship is worth different [27]. Then the tension process is simulated as an input ABAQUS of the real stress strain relationship after the necking, and the simulation results of different w values are compared with the tensile test data. The most reasonable w value is derived. Take the base metal, for example, from Figure 7: it can be known that $\varepsilon_{u}=0.29$ and $\sigma_{u}=410\;\mathrm{MPa}$.
(3)$$\sigma =\sigma_{u}\left[w\left(1+\varepsilon -\varepsilon_{u}\right)+\left(1-w\right)\left(\frac{\varepsilon^{\varepsilon_{u}}}{\varepsilon_{u}{}^{\varepsilon_{u}}}\right)\right]$$

As shown in Figure 8, The load displacement curve obtained from simulation is in good agreement with the experimental data when the weight coefficient w = 0.7. The weight of weld material is calculated as above. Therefore, a modified stress-strain curve is obtained by using the functional relation for the strip steel used in this paper.

## 4. Determination of XFEM Parameters

In this study, the finite element software ABAQUS was used to simulate the edge cracking behavior of the welded strip steel under the cold rolling process. The model is built by solid element with hexahedral structural mesh C3D8R in abaqus2020. The extended finite element model was adopted and the maximum principal stress (MAXPS) damage criterion was accepted as the crack propagation standard. The maximum principal stress $T_{0}$ and fracture energy $\Gamma_{0}$ are mainly two parameters in the MAXPS damage criterion. The two parameters were obtained through the hybrid technique which combines uniaxial tensile test, fracture toughness test and simulation [28], respectively.

## 5. Determination of the Maximum Principal Stress, $T_{0}$

The maximum principal stress was obtained by the hybrid technique [25]. Unstable fracture point which is the turning point at the stress-strain curve to decrease, was determined by the tensile test. The width of the necking section of the specimen was measured at the unstable point before rupture.

Under the pure elastoplastic condition, uniaxial tensile test simulation was carried out with the same boundary condition as the real test, and the maximum stress was obtained when the necking section width equals to the same value in the real test. The stress is the regarded as the maximum principal stress $T_{0}$ and the principle of the method is shown in Figure 9. The simulated stress value and the necking section width at the center of specimen are shown in Figure 10. It can be seen that the width of the necking section is 14 mm when the specimen is stretched to the unstable fracture point under the uniaxial tensile test. So when the width of the necking section equals to 14 mm under the numerical simulation condition, the maximum stress at the center of the section can be obtained as the maximum principal stress 543 MPa (in other words $T_{0}$ = 543 MPa).

Similarly, the maximum principal stress of weld metal was obtained by the same method and the maximum principal stress for the weld metal is about 713 MPa, as shown in Figure 11.

## 6. Determination of the Fracture Energy, Γ_0_

The fracture energy value is another important parameter for the XFEM simulation. The fracture energy of the base metal was measured directly by the essential work of fracture (EWF) test, and the fracture energy of the weld metal can only be measured by the spherical indentation method due to its narrow size.

## 7. Fracture Energy of Base Metal

In the process of fracture, the total energy required for fracture propagation per unit area is decomposed into basic fracture work $w_{e}$ and non-basic fracture work $w_{p}$. $w_{e}$ is the energy required to generate a new crack surface under the unit area, which acts on the fracture stage of the ligament. $w_{p}$ acts on the plastic deformation after the crack propagation. At the moment of crack initiation, the value of is equal to Ji, that is the initial crack toughness corresponding to the J-R resistance curve. After the crack extension, the $w_{e}$ and J-integral values are different. The EWF test is a recognized test method for obtaining the $w_{e}$ value. Essential work of fracture (EWF) is a very popular concept in evaluating the fracture toughness of thin plates. Abdellah [29] once used EWF test to measure the fracture toughness of thin aluminum plate, and the test results were used to verify the correctness of XFEM and J-integral simulation results. The results for the essential and non-essential fracture indicate good agreement with EWF fitting. The double edge-notched tensile (DENT) specimen was commonly used in EWF test and also adopted in this study. The schematic diagram of the DENT is shown in Figure 12.

The specimen under the load of P makes the ligament of the length *L* completely yield, and then the plastic crack extends along the ligament area until the specimen is completely broken.

The total work required for the fracture process of the specimen, $W_{f}$ by Equation (4) can be obtained as:(4)$$W_{f}=\int_{0}^{\delta_{c}}pd\delta$$
where $\delta_{c}$ is the line displacement of the specimen when the specimen is broken. The p is the experiment load. $W_{f}$ is the total work for fracture and is decomposed into two parts, one part is the basic work $W_{e}$, which is the energy density per unit area and the surface release work at the fracture of the ligament. The other part is the plastic work $W_{p}$, which is the volumetric energy of plastic deformation after the crack propagation and is proportional to $BL^{2}$. β is the plastic deformation shape factor.

$W_{f}$ can also be expressed as:(5)$$W_{f}=W_{e}+W_{p}$$
(6)$$W_{f}=W_{e}BL+\beta W_{P}L^{2}$$
(7)$$W_{f}=\frac{W_{f}}{BL}=W_{e}+\beta W_{P}L$$

Therefore,$\;W_{f}$ is a linear curve with the length *L* of the ligament area per unit area, and the intersection of the curve with the *Y* axis is $W_{e}$. The different $W_{f}$ values in the figure corresponding to the different lengths of the ligamentous region of the DENT specimen. Thus,$\;W_{f}$, the value of total fracture work per unit area in the ligamentous area can be achieved as the intercept in the y-coordinate by the linear regression, and $\beta W_{P}$ is the slope of the regression line.

Williams and Rink [30] describe the basic working background of the essential work of fracture (EWF) test method and its relationship to the J test. They proposed the basic fracture work test criteria, which require DENT specimens to be larger than twice the width and length of the ligament zone. Sahoo et al. [31] pointed out that the fracture toughness of the strip steel can be obtained by the basic fracture work test, which required the length of the ligament area between three and five times the thickness of the specimen and the 1/3 width of the specimen. Based on the above literature, the size of the specimen used in this study is as follows: the width of the specimen D = 25 mm, the length H = 120 mm, the length of the clamping end h = 30 mm, the thickness B = 2.6 mm, the original thickness of the plate, and the lengths of the 4 groups, L, were selected as 5, 6, 7, 8 mm, respectively. In addition, in order to prevent the effect of warpage on the load displacement curve of the specimen during the tensile test, the ligament area is at the center of the specimen, and the left and right gaps of the specimen are strictly symmetrical. Two side notched tensile specimen of the base material are shown in Figure 13.

Figure 14 shows the tensile load displacement curve of DENT specimen. The fracture toughness value is 620 N/mm, 689 N/mm, 766 N/mm, 923 N/mm for the ligament length *L* of 5 mm, 6 mm, 7 mm, 8 mm, respectively. As shown in Figure 15, when *L* = 0, the corresponding $W_{f}$ value (initial fracture toughness) is 130.6 N/mm.

## 8. Fracture Energy of Weld Metal and Accuracy Verification of Spherical Indentation Experiment

Due to the narrow size of the weld metal, it is difficult to measure fracture toughness by conventional EWF test. In this section, the fracture toughness of weldment materials was then obtained by the spherical indentation. Automatic spherical indentation test is a continuous cyclic process in which a spherical indenter is driven by a motor and pressed vertically into the surface of the material during the test. The load-displacement curve of the indenter is obtained and the tested material’s properties are obtained by using the relevant elastoplastic theory and empirical formulas. Many literatures have studied the measurement of fracture toughness by automatic spherical indentation [32,33,34,35].

Griffith [36] obtained the stress intensity factor of a penetrating crack with the crack length of 2a. The fracture toughness $K_{\mathrm{IC}}$ can be expressed as
(8)$$K_{\mathrm{IC}}=\sigma_{f}\sqrt{\pi a}$$
(9)$$\sigma_{f}=\sqrt{2EW_{f}/\pi a}$$

$\sigma_{f}$ is the far field tensile stress during fracture, which is perpendicular to the crack. a is the half of crack length. *E* is the elastic modulus. According to the generalized Griffith theory, its expression is
(10)$$K_{\mathrm{IC}}=\sqrt{2EW_{f}}$$

Although Equation (10) is derived from an infinite plate, it has been proved that the equation is a universal one. Assuming that the indentation deformation energy corresponding to the crack initiation at a critical depth $h^{*}$ during indentation, the energy absorbed per unit contact area in indentation test can be correlated with the fracture energy $W_{f}$. Since a crack has two sides, 2$W_{f}$ can be represented by Equation (11).
(11)$$2W_{f}=\underset{h\to h^{*}}{\lim}\int_{0}^{h}\frac{4F}{\pi d^{2}}dh$$
where *F* is applied load, *h* is indentation depth, *d* is projection diameter of contact area between indenter and material, $h^{*}$ is critical down pressure depth, and $2W_{f}$ is energy absorbed per unit area of two crack surfaces [26].
(12)$$M=\frac{\pi }{(\frac{4}{3}\pi {)}^{\frac{2}{3}}}f^{\frac{2}{3}}$$

Lemaitre [37] proposed the equivalent strain principle to correlate the damage variable *M* with the elastic modulus *E*.
(13)$$E_{D}=E\left(1-M\right)$$

Oliver and Pharr [38] considered that with the increase of indentation depth, the damage of the material under the indenter would also increase. The effective modulus of elasticity ($E_{D}$) could be expressed by indentation parameters.
(14)$$E_{D}=\frac{1-v^{2}}{\frac{2\sqrt{A_{c}}}{\sqrt{\pi }L}-\frac{1-v_{i}^{2\;}}{E_{i}}}$$

In the Equation (14), v and $v_{i}$ represent the Poisson’s ratio of material and indenter, $E_{i}$ is the elastic modulus of indenter, $A_{c}$ is the projection area of indenter and specimen contact, and L is the unloading stiffness of each unloading curve obtained from indentation test (that is, the unloading slope).

Andersson [39] deduced that the critical void ratio $f^{*}$ equals 0.25 for stable crack initiation and propagation. The critical damage variable $M^{*}$ and the critical elastic modulus $E_{D}{}^{*}$ can be calculated by introducing the value into Equation (12) and the simultaneous Equation (13). The critical down pressure depth $h^{*}$, calculated by the linear relationship between the fitting ln (h) and ln ($E_{D}$), can be substituted into Equation (14), and the fracture toughness value can be obtained.

The tungsten carbide head with a diameter of 0.1 mm was selected. The elastic modulus is 710 GPa and the Poisson’s ratio is 0.21. The automatic spherical indentation test cycle number was set to 8 times. The unloading ratio is 40% of the maximum load per cycle. The screw down rate of the head is 0.13 mm/min, the data acquisition rate of the testing machine is 200 Hz, and the testing temperature is 20 °C. In order to eliminate the influence of other factors and ensure the accuracy and repeatability of the experiment, 3 points of the sample was pressed. Figure 16a is the load depth curve fitted by the real-time test data of three automatic spherical indentation tests. Each curve corresponds to a test point. The coincidence degree of the three curves is high, especially test point 2 and test point 3.

For the load-depth curve obtained from the test, the slope of each unloading curve was obtained according to 8 unloading curves at 8 indentation depths, and the fitting curve of ln (h)-ln ($E_{D}$) was obtained. When the critical void fraction $f^{*}$ is 0.25, the initial *E* is 197 GPa. The corresponding critical elastic modulus $E_{D}{}^{*}\;$ is 102.4 GPa, and the critical indention depth $h^{*}$ is 0.0182 mm by linear fitting relation. The corresponding values are marked in Figure 17a. $W_{f}$ = 148 N/mm was calculated by Equation (11).

Sabita Ghosh et al. [40] highlights the applicability of BIT for evaluating flow behavior of engineering structural steels En24. The fracture toughness of En24 steel under different heat treatment conditions was calculated. These results are verified with the already established correlation. Sabita Ghosh et al. [40] evaluate the variation in mechanical properties due to heat treatment on as-received En steels. The data were verified by conventional test results. The spherical indentation technique is proved to be an effective technique for evaluating small mechanical properties.

In order to further verify the accuracy of fracture energy of weld material obtained by spherical indentation test, automatic spherical indentation test was carried out on base metal. The fracture energy obtained was compared with the fracture work data for confirmation. Take three test points on the base metal, and the load-indentation depth curve is shown in the Figure 16b. Obtain the slope of each unloading curve through the load-indentation depth curve and further obtain the fitting curve of ln (h) and ln ($E_{D}$). The initial elastic modulus *E* of the material is 220 GPa, and $E_{D}{}^{*}$ can be calculated to be 114 GPa.

Similar to the above, the fitting curve of ln (h)-ln ($E_{D}$) is obtained as shown in the Figure 17b. The critical indention depth $h^{*}$ is 0.0236 mm by linear fitting relation. The corresponding values are marked in figure. The fracture energy is 120.9 N/mm by Equation (11). The error of the base metal fracture energy of 130.6 N/mm obtained by the basic work of fracture test is 7.6%. Since the test error is within the allowable range, the spherical indentation test is applicable.

## 9. Verify the Maximum Principal Stress, $T_{0}$

In this section, the maximum principal stress, $T_{0}$, were verified by using the XFEM. The first simulation was performed on the thin-plate tensile specimen of base metal to check the accuracy of the value of $T_{0}$ shown in Figure 9 and Figure 10. In Figure 18a, the calculation using the XFEM as a dashed line fits the test load–displacement curve and unstable fracture point precisely when the $T_{0}$ is equal to the 543 MPa. This result shows that the maximum stress is sensitive to load–displacement simulation curve. In Figure 9a,b the loading displacement was 14 mm in the experiment process, and the same displacement value was loaded in the simulation. Through the comparison of the thin-plate tensile experiment and simulation with the XFEM, the maximum principal stress, $T_{0}$, which is equal to 543 MPa can satisfy the actual test requirement.

The same method was used to verify the maximum principal stress of weld material. As shown in Figure 18b. Through the comparison of tensile experiment and simulation with the XFEM, the maximum principal stress, $T_{0}$, which is equal to 713 MPa can satisfy the actual test requirement.

## 10. Finite Element Analysis for Cold Rolling Procedure

A four continuous rolling processes were studied in this paper. Since the size of heat affected zone (HAZ) is very small, the HAZ was omitted and only the weld and base metal were considered in the current finite element models. The preset edge flaw in the weld metal, base metal and the interface between the weld and base metal are, respectively, shown in Figure 19.

The model of the mill roller and the strip steel are shown in Figure 20a. The diameter of the roller R is 180 mm, and the sizes of the strip steel are L = 30 mm, W = 15 mm and B = 1.3 mm. Dynamic implicit algorithm was applied in the numerical analysis. Since the boundary conditions of the upper and lower rolls are symmetry, the symmetrical modeling along the thickness direction were used. The width of the sheet is also fixed to ensure the width of sheet keeps constant during the cold rolling process.

For rolling motion boundary conditions, the angular velocity applied to the roll in the counterclockwise direction with the value w = 5 rad/s and the initial velocity (v) of the strip steel along the rolling direction is 180 mm/s.

Since the elastic modulus of the mill roller is much higher than the elastic modulus of the strip steel, the roller was considered as an analysis rigid body in the finite element model. The strip steel was considered as an isotropic elastoplastic solid. The eight-node hexahedral solid element with reduction integration, C3D8R, was employed for the whole mesh. In this paper, the element near the notch tip was refined as shown in Figure 20b to satisfied the mesh independency requirement and ensure the result accuracy. There are more than 60,000 elements for the different models. The maximum principal stress criterion was adopted in the extended finite element model, and the parameters for the base and weld metal were obtained from the above-mentioned study.

## 11. Result and Discussion

Figure 21a,b shows experiment and simulation results of the behavior of a crack with the initial crack length is 5 mm on the base metal edge during multi-pass cold rolling. It shows that the direction and length of crack extension predicted by FEA are consistent with the experimental results as a whole.

The reduction of each rolling is 0.2 mm and the crack propagation behaviors of weld, base metal and the interface were compared. When the prefabricated crack is in the weld metal, the edge crack is at an angle of about 135° with the rolling direction, and extends along with the rolling process. The simulation result shown in the Figure 22 can be explained from the material surface hardening. When the left strip enters the roll first, plastic deformation occurs in the material, and the strength and hardness of the material increase, preventing the crack from spreading to the left.

Figure 23 and Figure 24 show both crack propagation when preset edge flaws were at the base metal 1(BM1) and interface between the weld and BM1. This result is the same as the case of the steel sheet edge crack. D.Q.ZAN [25] used the cohesion zone model to predict the edge crack propagation of steel sheet in the cold rolling process. The final result is that the crack spreads 45° along the rolling direction. The front side of the edge flaw first contact with the roller and the stress concentration appears in the area along the angle about 45° from the rolling direction near the front side. In addition, sine the right end of the crack is the weld zone, the fracture energy of the weld is much larger than that of the base metal, so the crack is also easier to expand to the base metal. In the end, the edge crack occurs at an angle of 45° from the rolling direction when the crack extends into base metal instead of weld metal.

In the above study, the prefabricated cracks located at the base metal and the interface that entered the rolling area first. In order to conduct a complete study of the whole strip steel rolling, the following prefabricated cracks are set at the BM2. The initial crack was prefabricated at a distance of 2.5 mm and 0.5 mm from the weld, and the crack propagation was observed after a single rolling process.

As shown in Figure 25, Prefabricated initial crack at a distance of 2.5 mm from weld propagated toward the direction of 45°, also since the driving force at the trailing edge of the crack was smaller than that at the leading edge, which is less affected by the welded zone. In addition, prefabricated initial crack at a distance of 0.5 mm from weld was affected since it was too close to the weld. The crack expanded toward the direction of 135° and extended to the base metal. This is also due to the fact that the fracture energy of the weld is too large, resulting in cracks more easily extend to the base material area.

The crack propagation length of four different edge flaws are shown in Figure 26. location 1 refers to the preset flaw located in BM1, location 2 refers to preset flaw located in the interface between WM and BM1, location 3 refers to the preset flaw located in WM and location 4 refers to the preset flaw which is located in BM2, respectively.

As shown in Figure 26, the crack propagation length at the BM2 is the largest, followed by the BM1 crack, the interface crack and the weld crack. Since the base metal has the smallest fracture energy value and the least resistance to crack propagation, it has the longest propagation length. After the first rolling, the crack propagation length at the BM2 and at the WM is, respectively, 1.2 mm and 0.9 mm. However, after the second rolling, the crack propagation length at the BM2 is twice the crack propagation length at the WM, and after the last rolling, it is four times. The gap in the length of crack propagation is increasing. Quan Sun [13] have verified it in previous research.

It should be noted that BM2 enters the roll later than BM1, the crack propagation of BM2 is the most obvious. The welding seam has the largest fracture energy and the strongest resistance to crack propagation, so the propagation length is the shortest. The material properties at the interface are between them, so the crack propagation length is also between them.

## 12. Conclusions

In this paper, the initiation and propagation of the edge cracking were investigated under the four continuous passes rolling process by using the XFEM. The XFEM parameters were obtained from a hybrid technique and fracture toughness test. By prefabricating the initial crack at different locations, the characteristics of the edge crack initiation and propagation were analyzed. The following conclusions can be obtained:(1)The results of the simulation cold rolling show that the edge crack occurs at an angle of 45° or 135° from the rolling direction and extends along the direction towards base metal due to the different material properties.(2)For the materials used in this study, the fracture resistance of base metal is worse than that of weld metal. The growth and propagation rate of crack in the base area is greater than that in the weld metal under continuous rolling process. This means that more caution should be paid on the side defect at the base metal than at the weld metal.(3)If the cracks at BM1 and BM2 are far away from the weld, then the cracks will spread towards the base material area rather than towards the weld. However, the crack growth of BM2 is the most severe, and BM2 is more prone to fracture.(4)For cold-rolled sheet with weld, there are many defects. In this paper, only the initial defects at the edge are studied to a certain extent. In addition, there are wave defects, warping defects, inclusion defects, etc. The subsequent research can consider the comprehensive influence of these plate defects and edge defects.

## Figures and Tables

**Figure 1 materials-16-05870-f001:**
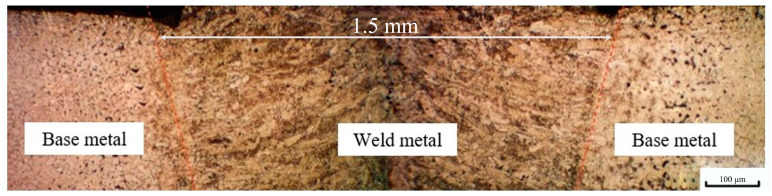
Metallographic of the cross section of the sheet.

**Figure 2 materials-16-05870-f002:**
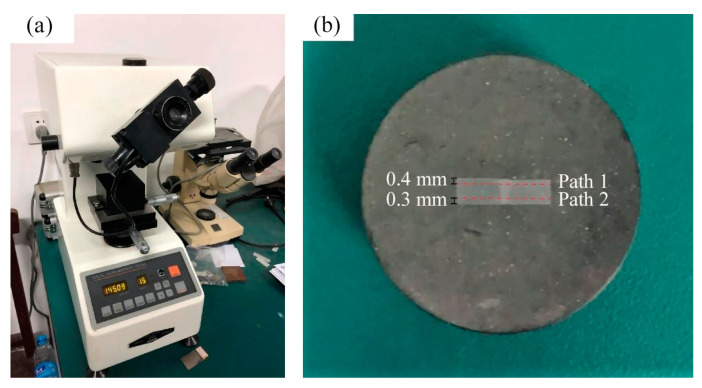
(**a**) Vickers microhardness tester and (**b**) inlaid sample with welds.

**Figure 3 materials-16-05870-f003:**
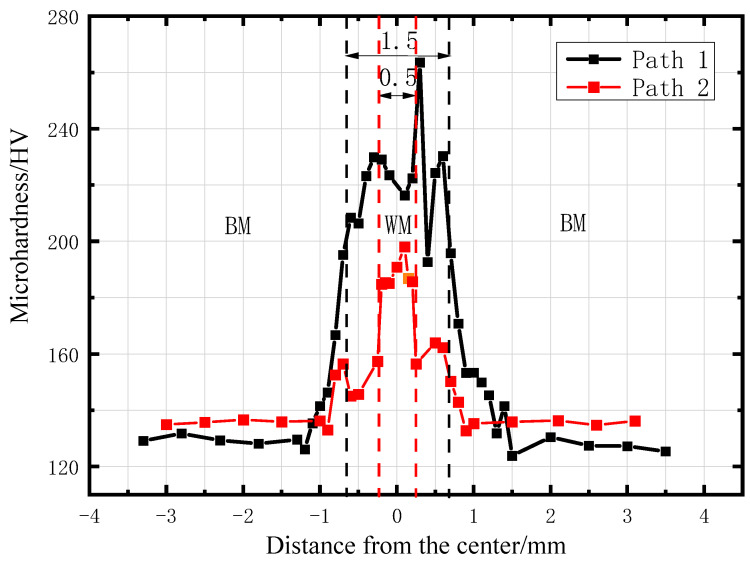
Microhardness profile on Path1 and Path2.

**Figure 4 materials-16-05870-f004:**
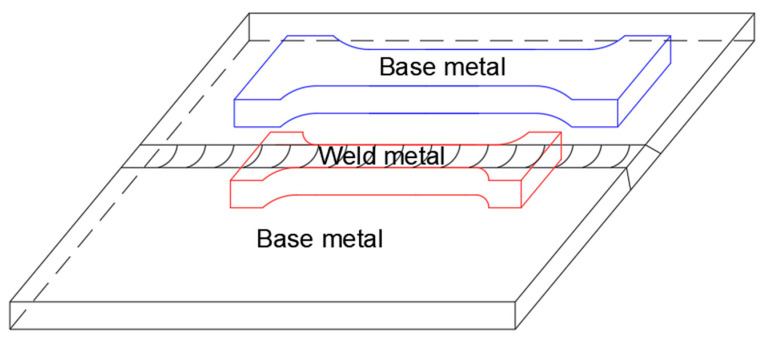
Schematic drawing of tensile specimen.

**Figure 5 materials-16-05870-f005:**
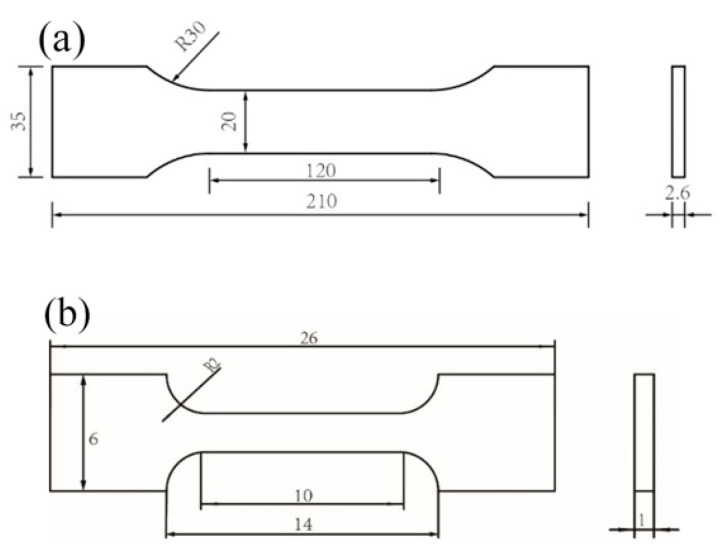
The size diagram of specimen: (**a**) base metal, (**b**) weld metal.

**Figure 6 materials-16-05870-f006:**
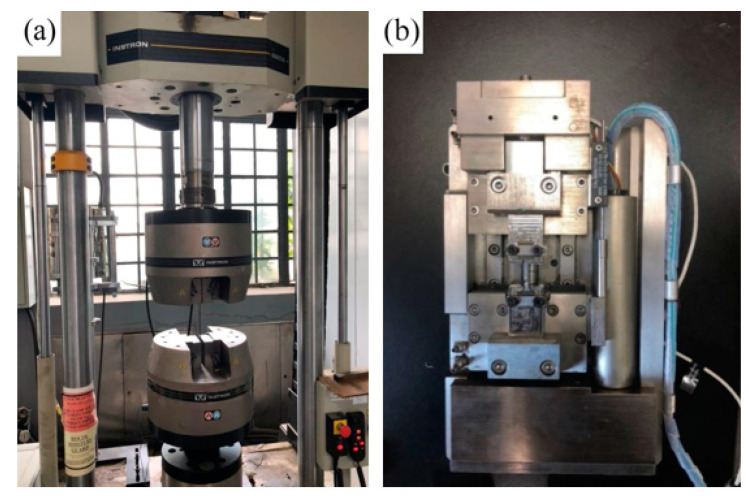
Specimen and testing machine for base metal and weld metal.

**Figure 7 materials-16-05870-f007:**
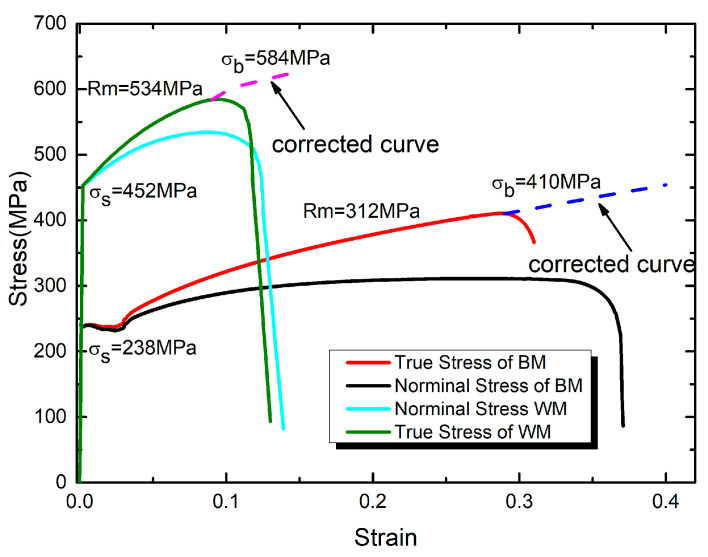
Stress strain curve of base metal and weld material.

**Figure 8 materials-16-05870-f008:**
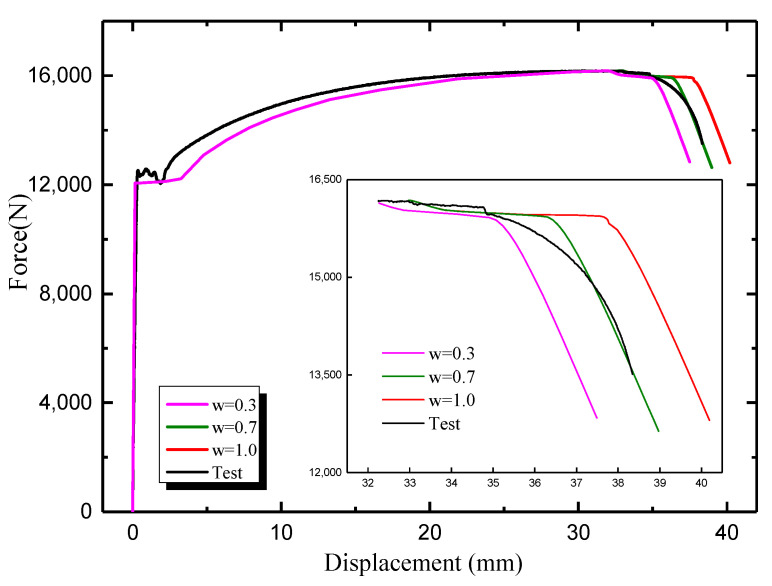
Comparisons between simulated load-displacement curves and experimental curves under different weights.

**Figure 9 materials-16-05870-f009:**
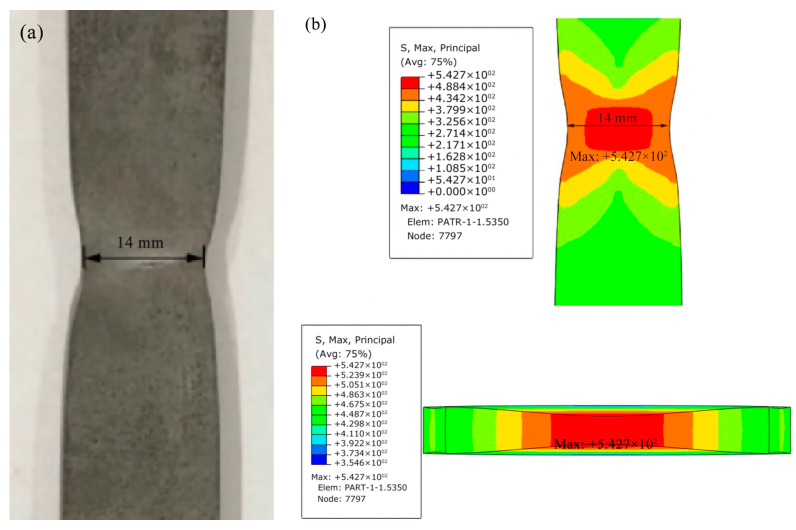
Determination of maximum principal stress, T_0_, using the hybrid technique: Tensile test specimen at onset of unstable fracture when the width decreasing to 14 mm, and el-pl FEM simulation result of the instant necking section: (**a**) Tensile test specimen and (**b**) Simulation result of the instant necking section.

**Figure 10 materials-16-05870-f010:**
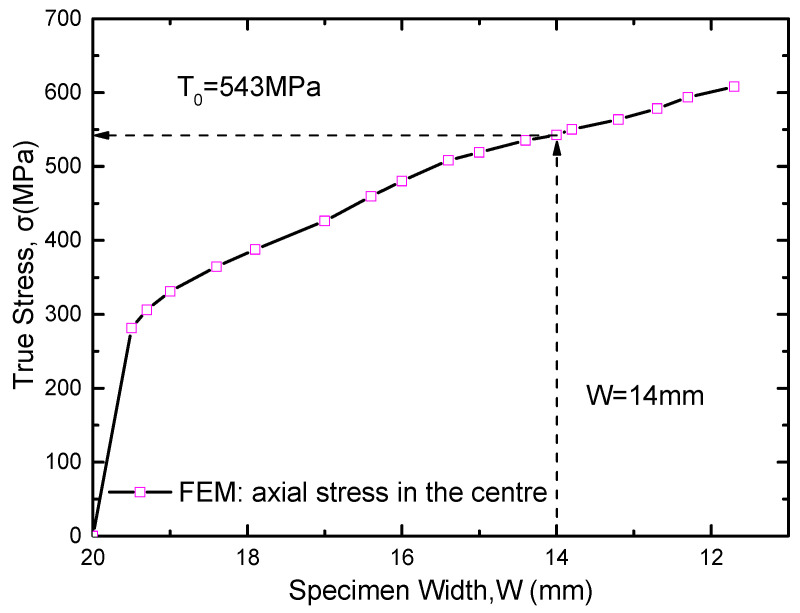
The curve of the stress in the center of the necking section vs. specimen width from the initial to the final.

**Figure 11 materials-16-05870-f011:**
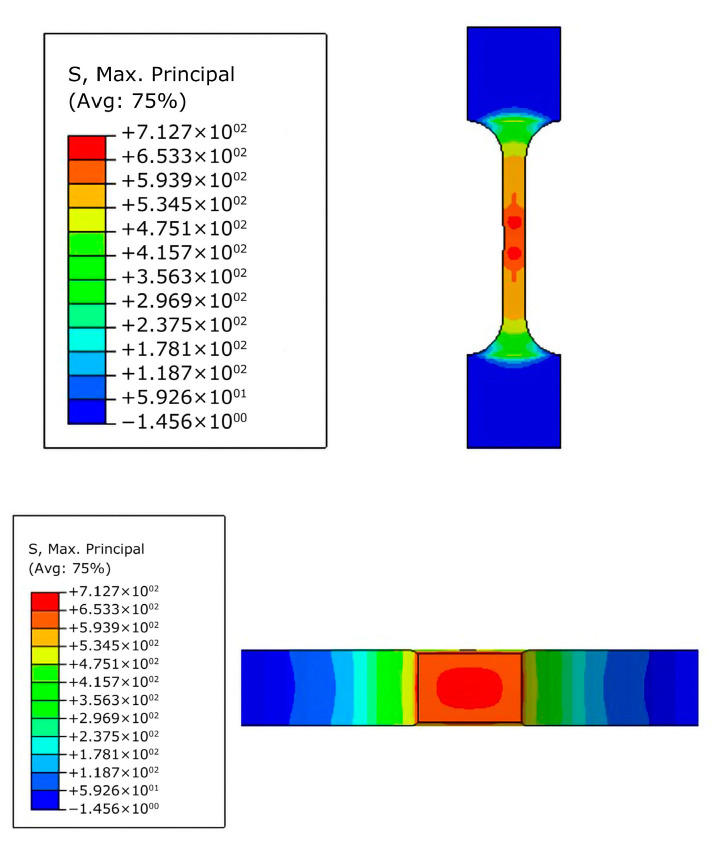
Determination of maximum principal stress, T_0_, of the weld metal.

**Figure 12 materials-16-05870-f012:**
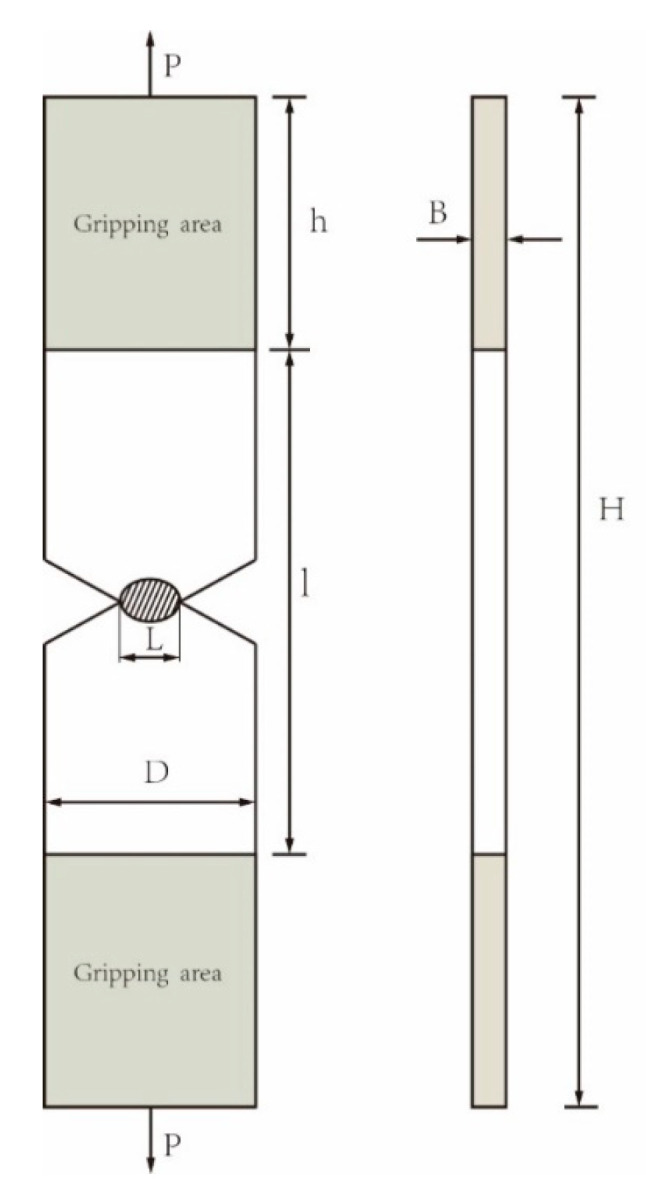
Double edge-notched tensile (DENT) specimen.

**Figure 13 materials-16-05870-f013:**
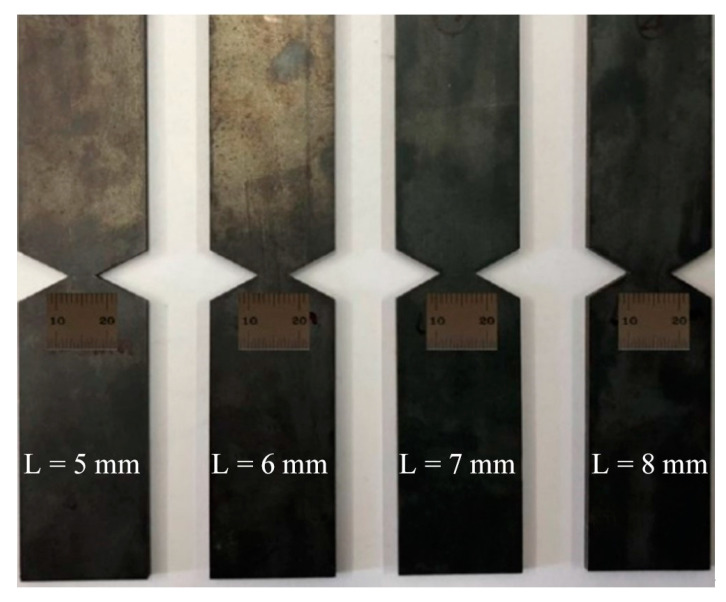
DENT specimens.

**Figure 14 materials-16-05870-f014:**
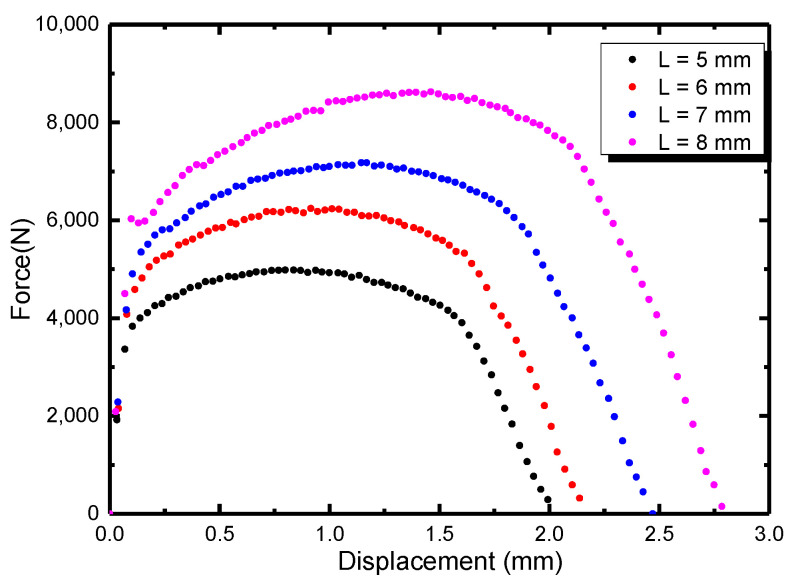
The load displacement curve of DENT specimen.

**Figure 15 materials-16-05870-f015:**
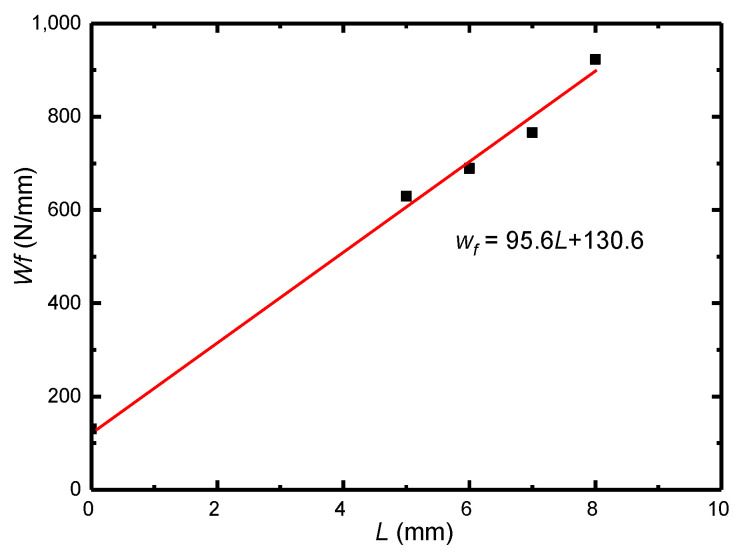
EWF test linear fitting curve.

**Figure 16 materials-16-05870-f016:**
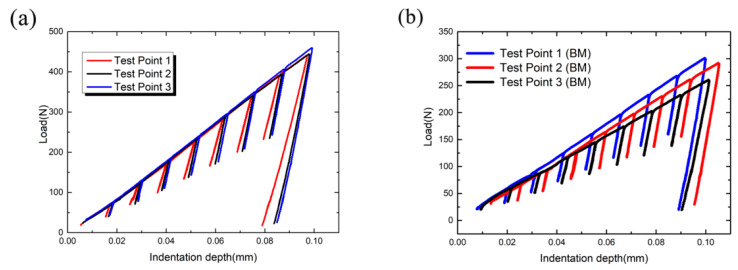
Load indentation depth curve of automatic spherical indentation test: (**a**) weld metal and (**b**) base metal.

**Figure 17 materials-16-05870-f017:**
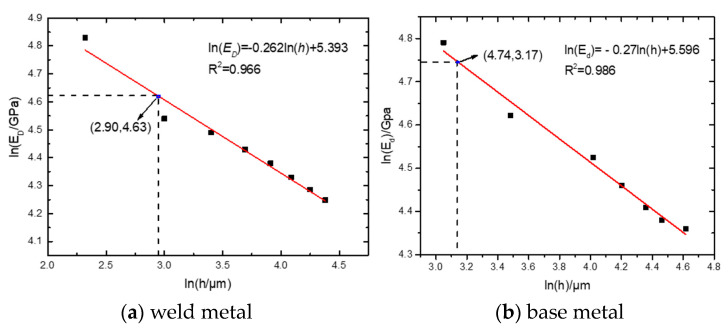
Linear fitting relationship between ln (h) and ln ($E_{D}$).

**Figure 18 materials-16-05870-f018:**
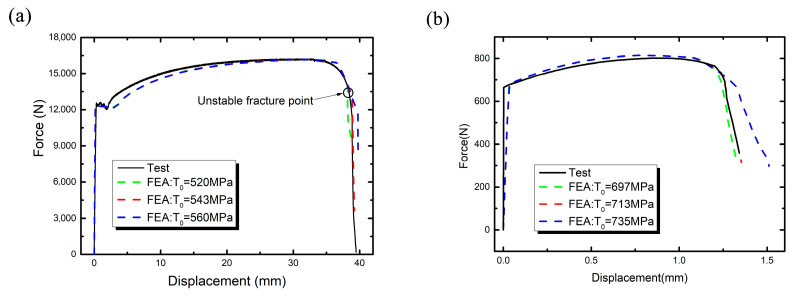
Simulation of the tensile load–displacement curve vs. experiment: (**a**) base metal, (**b**) weld metal.

**Figure 19 materials-16-05870-f019:**
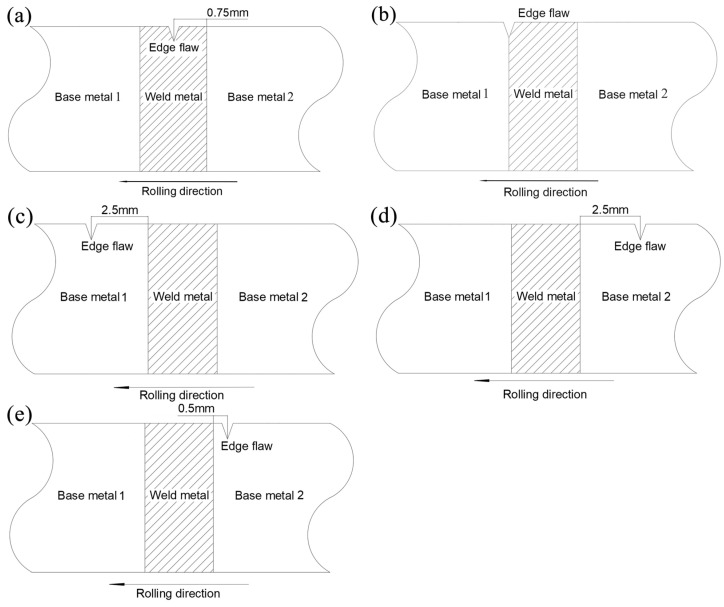
The location of the preset edge flaws: (**a**)in WM, (**b**)in the interface, (**c**) in BM1 2.5mm from left interface, (**d**) in BM2 2.5mm from right interface, (**e**) in BM2 0.5mm from right interface.

**Figure 20 materials-16-05870-f020:**
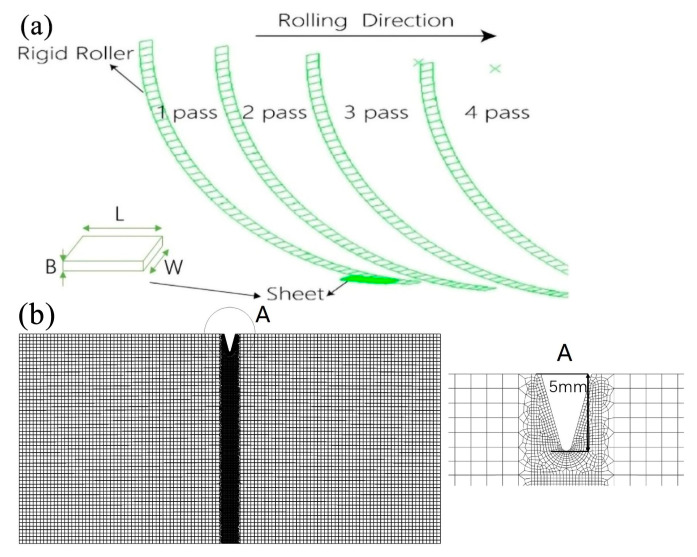
Finite element model of cold rolling process: (**a**) the model of the mill roller and strip steel, (**b**) the schematic diagram of meshing.

**Figure 21 materials-16-05870-f021:**
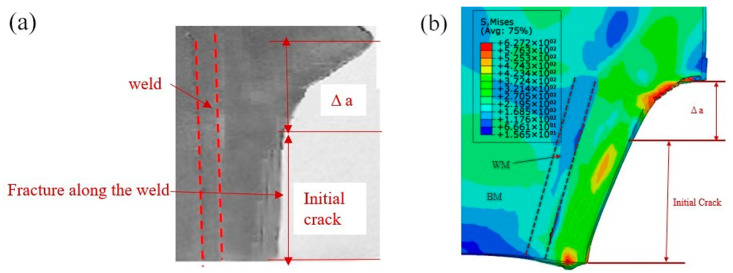
Comparison of experiment and simulation results: (**a**) experiment result, (**b**) simulation result.

**Figure 22 materials-16-05870-f022:**
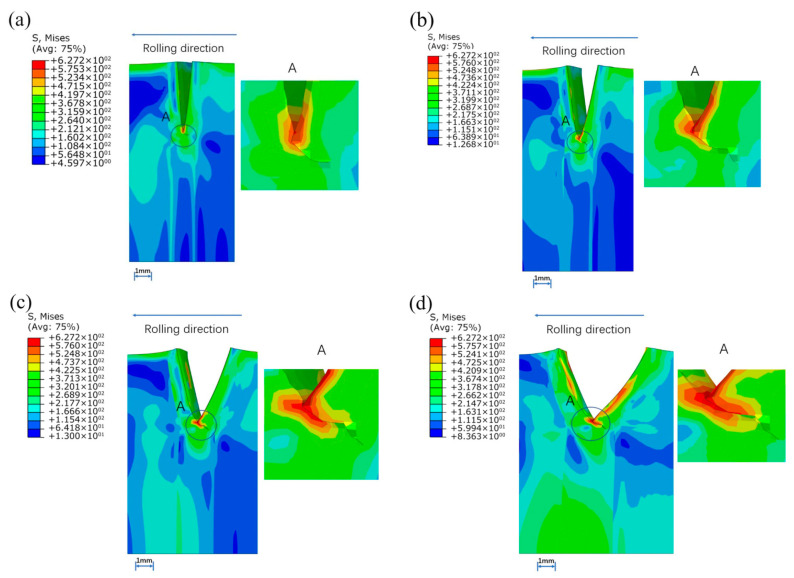
Crack propagation when preset crack was set in the weld metal: (**a**) 1st pass, (**b**) 2nd pass, (**c**) 3rd pass, (**d**) 4th pass.

**Figure 23 materials-16-05870-f023:**
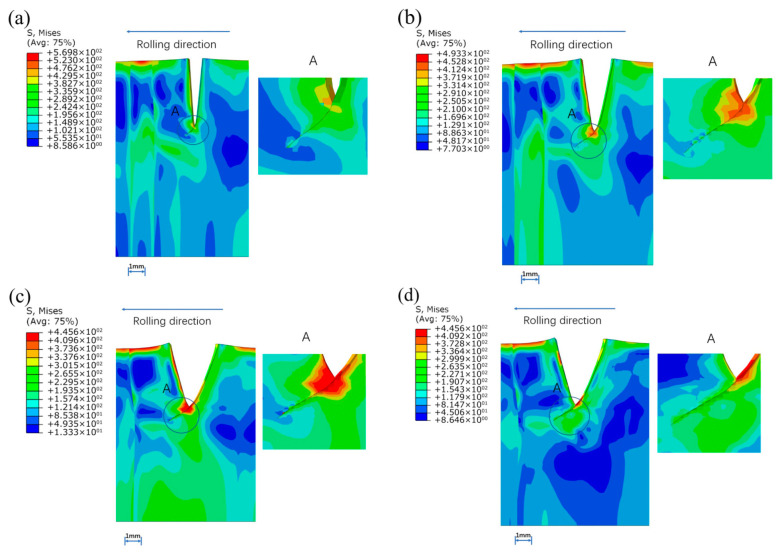
Crack propagation when preset crack was set in the BM1: (**a**) 1st pass, (**b**) 2nd pass, (**c**) 3rd pass, (**d**) 4th pass.

**Figure 24 materials-16-05870-f024:**
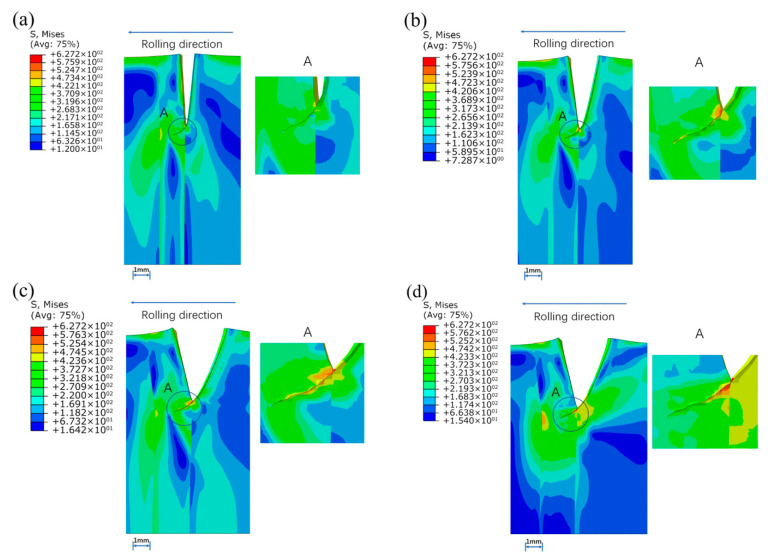
Crack propagation when preset crack was set at the interface of the weld metal and BM1: (**a**) 1st pass, (**b**) 2nd pass, (**c**) 3rd pass, (**d**) 4th pass.

**Figure 25 materials-16-05870-f025:**
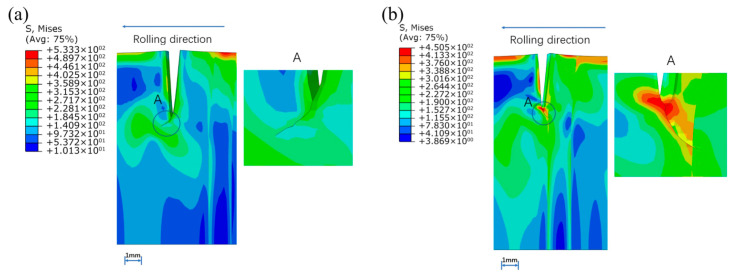
Crack propagation when preset crack was set at the BM2: (**a**) 2.5mm from the right interface, (**b**) 0.5mm from the right interface.

**Figure 26 materials-16-05870-f026:**
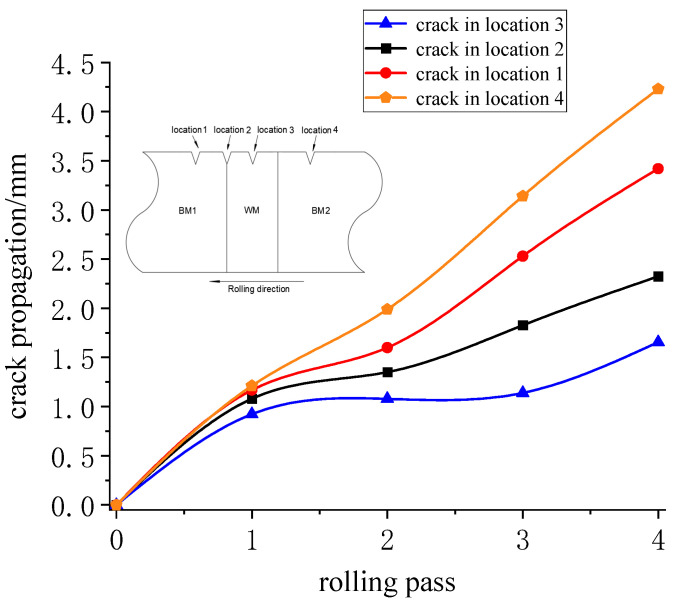
Crack propagation in the 4 pass rolling.

**Table 1 materials-16-05870-t001:** Chemical analysis of the normalizing cold rolling steel (% in weight).

Si	P	AI	S	Ti	Mn	C
0.51	0.682	3	0.04	0.018	0.347	0.016

## Data Availability

The data that support the findings of this study are available from the corresponding author upon reasonable request.

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
