# Peer review of "Application of Extended Finite Element Method for Studying Crack Propagation of Welded Strip Steel in the Cold Rolling Process"

_materials, 2023, doi:10.3390/ma16175870_

Round 1
Reviewer 1 Report
The manuscript entitled "Application of XFEM for studying crack propagation of welded strip steel in the cold rolling process" applied the Extended Finite Element Method (XFEM) to evaluate the crack propagation in welded strip steel during the cold rolling process. This manuscript was evaluated and major revisions are required.
1. The important results of the present work must be added in the abstract as numbers or percentages.
2. The introduction section required modification by adding more moderate studies about the subject. It was noted that only references No. 1 and 19 are moderates (the year 2021(.
3. For the cold rolling process, it is required more details to show the reason that the damage in the edge of the strip steel is higher than that in the middle area.
4. The novelty of this work needed more details to be clear to the readers.
5. On page 3 (Table 1), it is required the standard method and used device of the chemical elements analysis of the silicon strip steel sheet.
6. On page 3 (line 126), further details are required about the applied Path1 and Path2.
7. The manuscript in the results and discussion section doesn't answer deeply the importance of using the XFEM for studying crack propagation of welded strip steel in the cold rolling process. More discussions and details are required to explain the results in Figures 21, 22, 23, and 24.
8. Also, a more discussion detailed is required for Figure 25 to descript the difference between the crack propagation in the four-pass rolling.
9. It is required to compare the present work result with the results of other studies.
10. Rearrange the conclusion section by focusing on the important findings of the present work in one paragraph.
11. Recent references are required to cover the manuscript topics in all sections.
Author Response
The places you mentioned that need to be modified have been modified in the paper and marked in yellow.

Reviewer 2 Report
The paper titled, “Application of XFEM for studying crack propagation of welded 1
strip steel in the cold rolling process” was submitted by Jianjun CHEN and coauthors. In my opinion, a major revision is needed to accommodate the high-quality requirements of the Journal.
1. It is necessary that point out the innovation of your work in the abstract clearly. Please re-write the abstract and specify the optimization done with this method by mentioning the quantity.
2. Please include reference 1 inside the manuscript in the Introduction section.
3. The introduction part should be improved - focus on recent advances in this area rather than on common things that are known to everyone.
4. Please improve the quality of the Figures, especially Figure 6, 7, and 8, which should be replaced with a more suitable Figure.
5. Please compare your work with similar works.
The paper titled, “Application of XFEM for studying crack propagation of welded 1
strip steel in the cold rolling process” was submitted by Jianjun CHEN and coauthors. In my opinion, a major revision is needed to accommodate the high-quality requirements of the Journal.
1. It is necessary that point out the innovation of your work in the abstract clearly. Please re-write the abstract and specify the optimization done with this method by mentioning the quantity.
2. Please include reference 1 inside the manuscript in the Introduction section.
3. The introduction part should be improved - focus on recent advances in this area rather than on common things that are known to everyone.
4. Please improve the quality of the Figures, especially Figure 6, 7, and 8, which should be replaced with a more suitable Figure.
5. Please compare your work with similar works.
Author Response

(The authors gave the same response as above.)

Reviewer 3 Report
In the manuscript entitled “Application of XFEM for studying crack propagation of welded strip steel in the cold rolling process”, Chen et al. employed the Extended Finite Element Method (XFEM) to analyze the crack propagation behavior in welded strip steel during cold rolling. The authors conducted tensile testing, essential work of fracture (EWF) testing, hardness testing, and elastoplastic finite element simulations to evaluate the maximum principal stress and fracture energy utilized in XFEM for the base metal and weld metal respectively. Additionally, the authors used a continuous cold rolling model to investigate the crack propagation behaviors in the base metal, weld metal, and interface region.
Overall, some issues are associated with this research article, which need to be addressed before possible publication.
Please find the attached annotated file to see my comments.
Lastly, I would like to say Materials Journal publishes high-quality research articles related to welding. Based on my comments mentioned in the annotated file, the recommendation is Major Revision.

Author Response

(The authors gave the same response as above.)

Round 2
Reviewer 1 Report
Many thanks to the authors for completing the required revisions.
Best regards
Reviewer 2 Report
It can be accepted.